# Increase in the Expression of Glucose Transporter 2 (GLUT2) on the Peripheral Blood Insulin-Producing Cells (PB-IPC) in Type 1 Diabetic Patients after Receiving Stem Cell Educator Therapy

**DOI:** 10.3390/ijms25158337

**Published:** 2024-07-30

**Authors:** Yong Zhao, Boris Veysman, Kristine Antolijao, Yelu Zhao, Yldalina Papagni, Honglan Wang, Robin Ross, Terri Tibbot, Darinka Povrzenic, Richard Fox

**Affiliations:** 1Throne Biotechnologies, Paramus, NJ 07652, USA; 2Fresenius Medical Care North America, Waltham, MA 02451, USA; 3Life Line Stem Cell Tissue, Cord Blood Bank, New Haven, IN 46774, USA

**Keywords:** GLUT2, peripheral blood insulin-producing cells, cord blood-derived stem cells, stem cell educator therapy, type 1 diabetes, autoimmune diseases, IL-1β

## Abstract

Multicenter international clinical trials demonstrated the clinical safety and efficacy by using stem cell educator therapy to treat type 1 diabetes (T1D) and other autoimmune diseases. Previous studies characterized the peripheral blood insulin-producing cells (PB-IPC) from healthy donors with high potential to give rise to insulin-producing cells. PB-IPC displayed the molecular marker glucose transporter 2 (GLUT2), contributing to the glucose transport and sensing. To improve the clinical efficacy of stem cell educator therapy in the restoration of islet β-cell function, we explored the GLUT2 expression on PB-IPC in recent onset and longstanding T1D patients. In the Food and Drug Administration (FDA)-approved phase 2 clinical studies, patients received one treatment with the stem cell educator therapy. Peripheral blood mononuclear cells (PBMC) were isolated for flow cytometry analysis of PB-IPC and other immune markers before and after the treatment with stem cell educator therapy. Flow cytometry revealed that both recent onset and longstanding T1D patients displayed very low levels of GLUT2 on PB-IPC. After the treatment with stem cell educator therapy, the percentages of GLUT2^+^CD45RO^+^ PB-IPC were markedly increased in these T1D subjects. Notably, we found that T1D patients shared common clinical features with patients with other autoimmune and inflammation-associated diseases, such as displaying low or no expression of GLUT2 on PB-IPC at baseline and exhibiting a high profile of the inflammatory cytokine interleukin (IL)-1β. Flow cytometry demonstrated that their GLUT2 expressions on PB-IPC were also markedly upregulated, and the levels of IL-1β-positive cells were significantly downregulated after the treatment with stem cell educator therapy. Stem cell educator therapy could upregulate the GLUT2 expression on PB-IPC and restore their function in T1D patients, leading to the improvement of clinical outcomes. The clinical data advances current understanding about the molecular mechanisms underlying the stem cell educator therapy, which can be expanded to treat patients with other autoimmune and inflammation-associated diseases.

## 1. Introduction

Type 1 diabetes (T1D) is one of the serious autoimmune diseases that causes a deficit of pancreatic islet β cells. Millions of individuals worldwide have T1D, and its incidence has increased post-COVID-19 pandemic [1,2,3,4]. T1D patients require lifelong management with daily glucose monitoring and insulin injections. While daily insulin injections offer limited control over their blood sugar levels and may delay the onset of chronic complications (e.g., cardiovascular diseases, retinopathy, neuropathy, and kidney disease) [5,6,7,8,9,10], a true cure has proven elusive despite intensive research efforts over the past decades [11,12,13,14]. Recent clinical trials have highlighted the limits of conventional immune and stem cell therapy [15,16,17,18] and underscored the need for novel approaches that not only overcome multiple immune dysfunctions but also help restore islet β cells. To address these two key issues, we have developed a unique and novel procedure designated the stem cell educator (SCE) therapy [19,20]. With this technology, a patient’s blood is circulated through a blood cell separator, wherein the patient’s immune cells are co-cultured with adherent cord-blood-derived multipotent stem cells (CB-SC) in vitro, after which “educated” immune cells are infused into the patient’s circulation. Over the last 12 years, our unique technology has been evaluated through international multi-center clinical studies, which have demonstrated its clinical efficacy and safety to treat patients with T1D [19,21], type 2 diabetes [22], alopecia areata [23], and other autoimmune diseases. Recently, stem cell educator therapy has been approved by the United States FDA for Regenerative Medicine Advanced Therapy (RMAT) designation. Notably, stem cell educator therapy is the only therapy to date to safely and efficiently correct autoimmunity and restore β cell function in T1D patients.

Previous work characterized a novel cell population from adult human blood, designated peripheral blood insulin-producing cells (PB-IPC) [24]. PB-IPC displayed characteristics of islet β-cell progenitors, including the expression of β-cell-specific insulin gene-associated transcription factors (e.g., MAFA and NKX6.1) and prohormone convertases PC1/3 and PC2, production of insulin, the ability to reduce hyperglycemia, and the ability to migrate into pancreatic islets after transplantation into the streptozotocin (STZ)-induced diabetic mice [24]. Our recent studies demonstrated that PB-IPC displayed a unique phenotype such as Lin1^−^CD34^−^CD45^+^CD45RO^+^CCR7^+^SOX2^+^OCT3/4^+^MAFA^+^GLUT2^+^ [25]. Notably, the differentiation potentials of PB-IPC were markedly increased after the treatment with platelet-derived mitochondria, giving rise to three-germ layer-derived cells such as neuronal cells, retinal pigmented epithelial (RPE) cells, and different lineages of blood cells [25,26].

Glucose, as the main source of cellular energy, enters cells through a group of membrane proteins designated glucose transporters (GLUT 1-12). GLUT2 is widely expressed in the central nervous system, liver, intestine, kidney, and pancreatic islet β cells, contributing to glucose transport and glucose sensing [27,28]. GLUT2 provides relatively high capability of transporting glucose through the cellular membrane [29,30,31]. To improve the clinical efficacy of stem cell educator therapy in the restoration of islet β cells, we analyzed the phenotype of PB-IPC in T1D patients before and after the treatment with stem cell educator therapy. Notably, we found the level of GLUT2 expression on PB-IPC was markedly increased in recent onset T1D subjects after receiving stem cell educator therapy. We further explored the therapeutic effects of stem cell educator therapy to improve GLUT2 expressions of PB-IPC in patients with longstanding T1D and other autoimmune- or inflammation-associated diseases.

## 2. Results

### 2.1. Low Expression of GLUT2 on Recent Onset T1D Patient-Derived PB-IPC

Using the previously established cellular markers [25], we characterized the PB-IPC from the peripheral blood of healthy donors (Figure 1A). To explore whether the development of T1D was associated with PB-IPC, we initially recruited recent onset T1D subjects (n = 9, aged from 11 to 38 years old, 3 females and 6 males) and examined the levels of their PB-IPC. Flow cytometry analysis revealed that the percentages of their GLUT2^+^ PB-IPC were at 4.79% ± 6.93% and much lower than the 59.94% ± 20.34% healthy control (n = 12, aged from 11 to 63 years old) (Figure 1B, *p* < 0.0001). The data suggested that PB-IPC was affected in T1D patients.

### 2.2. Upregulate the Expression of GLUT2 on Recent Onset T1D Patient-Derived PB-IPC

To explore the therapeutic effects of stem cell educator therapy on PB-IPC, the percentage of glucose transporter 2 (GLUT2) + PB-IPC was monitored in type 1 diabetic subjects before and after the treatment with stem cell educator therapy. Flow cytometry analysis revealed that the percentage of GLUT2^+^CD45RO^+^ PB-IPC was markedly increased after the treatment with stem cell educator therapy for one month in recent onset T1D subjects (*p* = 0.002, n = 10, Figure 2A,B). Using human C-peptide (a by-product of insulin biosynthesis) as an indicator for the restoration of endogenous islet β cell function, the levels of C-peptide were monitored after the treatment with stem cell educator therapy in these recent onset T1D subjects. Clinical data demonstrated that T1D subjects markedly increased their C-peptide levels relative to their baseline levels (0.87 ± 0.36 ng/mL) in the first month post-treatment with stem cell educator therapy (1.18 ± 0.49 ng/mL, *p* = 0.005), with the improved sugar control and the reduction of hemoglobin A1c (HbA1C) and total insulin dosages. Some of these T1D subjects stopped the external insulin.

Next, we tested the level of GLUT2^+^ PB-IPC in a prediabetic patient, which displayed high incidence to develop T1D with 3-positive T1D-associated autoantibodies such as glutamic acid decarboxylase 65 (GAD-65), zinc transporter 8 (ZnT8), and islet cell antibody (ICA) [32,33]. Flow cytometry demonstrated that this prediabetic subject displayed a low percentage of GLUT2^+^ PB-IPC at 7.64%. After receiving stem cell educator therapy, the percentage of GLUT2^+^ PB-IPC increased to 60.64% at the first month follow-up and 94.6% at the third month follow-up (Figure 2C). This subject maintained normal sugar control with HbA1c at 5.2% during the follow-ups. It suggests that stem cell educator therapy may prevent the development of T1D.

### 2.3. Upregulate the Expression of GLUT2 on Longstanding T1D Patient-Derived PB-IPC after Receiving Stem Cell Educator Therapy

To explore the level of GLUT2 expression in longstanding T1D patient-derived PB-IPC, flow cytometry analysis demonstrated that there were also low percentages of GLUT2^+^CD45RO^+^ PB-IPC at 9.07% ± 10.08% in longstanding T1D patient-derived PB-IPC (n = 8). After treatment with stem cell educator therapy, the percentages of GLUT2^+^CD45RO^+^ PB-IPC were markedly improved to 34.58% ± 17.28% relative to their baseline levels (Figure 3A, *p* = 0.018). Notably, subject HU2003 had been diagnosed with T1D for 58 years since age 1 year old and yet increased the percentages of GLUT2^+^CD45RO^+^ PB-IPC from 0% at baseline to 72.09% at the sixth month and to 89.89% at the twelfth month follow-up post the treatment with stem cell educator therapy (Figure 3B). The daily dosage of total insulin was reduced by about 30–40% with improved sugar control.

Additionally, subject HU2073 having T1D for 50 years increased the percentage of GLUT2^+^CD45RO^+^ PB-IPC from 2.14% at baseline to 71.1% at the first month follow-up, which may potentially contribute to the marked improvement in the neuropathy of his foot after receiving stem cell educator therapy.

### 2.4. Increase in the Expression of GLUT2 on PB-IPC in Patients with Other Autoimmune- or Inflammation-Associated Diseases after Receiving Stem Cell Educator Therapy

To determine whether the low expression of GLUT2 on PB-IPC was specific for T1D, we examined the levels of GLUT2 expression in patients with other autoimmune diseases (e.g., alopecia areata, Sjogren’s syndrome, and rheumatoid arthritis) and inflammation-associated neuronal degenerative disease such as Parkinson’s diseases (n = 3). Flow cytometry revealed very low expressions of GLUT2 on PB-IPC from these subjects before receiving the stem cell educator therapy. Notably, 7/8 of AA subjects markedly increased their percentages of GLUT2 expression from 0.25% ± 0.44% at baseline to 39.85% ± 24.43% at 3-month follow-up after receiving stem cell educator therapy (*p* = 0.004, Figure 4A,C). Additionally, flow cytometry analysis demonstrated that the levels of GLUT2 expressions were significantly upregulated on PB-IPC from 7.69% ± 6.26% at the baseline to 61.3% ± 14.52% at the first month follow-up in patients (n = 8) with other autoimmune- or inflammation-associated diseases after receiving stem cell educator therapy (Figure 4B,C). The clinical data indicates that the low expression of GLUT2 on PB-IPC might be the common marker for patients with T1D and other autoimmune- or inflammatory-associated diseases.

### 2.5. Downregulation of the Percentages of Inflamatory Interleukin (IL)1-β-Positive Cells in T1D Subjects after the SCE Therapy

Next, we explored whether chronic inflammation contributed to the downregulation of GLUT2 expression on PB-IPC in these autoimmune- or inflammation-associated diseases. Current T1D clinical studies showed that 6/10 of recent onset T1D subjects displayed high percentages of IL-1β-positive cells at 12.36% ± 5.82% before treatment. After treatment with stem cell educator therapy, they were markedly declined to 4.4% ± 3.85% (*p* = 0.028, Figure 5A). There were no significant differences for other cytokines, such as interferon γ (IFNγ, 0.053% ± 0.04 at baseline vs. 0.067% ± 0.11 at first month follow-up, *p* = 0.46), tumor necrosis factor alpha (TNFα) (0.069% ± 0.07 at baseline vs. 0.12% ± 0.18 at first month follow-up, *p* = 0.64), IL-13 (0.074% ± 0.06 at baseline vs. 0.307% ± 0.46 at first month follow-up, *p* = 0.78), and IL-17A (0.047% ± 0.08 at baseline vs. 0.06% ± 0.11 at first month follow-up, *p* = 0.74). Additionally, we explored other monocyte/macrophage-associated inflammatory chemokines [34,35] such as CC Motif Chemokine Ligand 2 (CCL2, also named Monocyte chemoattractant protein-1 (MCP-1)), CCL3 (macrophage inflammatory protein-1α (MIP-1α)), CCL4 (MIP-1β), C-X-C motif ligand 9 (CXCL9), and CXCL10 (IP-10). The levels of their expressions were very low and failed to show marked differences in these subjects before and after the treatment with stem cell educator therapy. It suggests that the monocyte/macrophage-derived inflammatory cytokine IL-1β might play an important role in T1D.

Additionally, 8/11 of subjects with other autoimmune- or inflammation-associated diseases also exhibited high levels of IL-1β-positive cells at 15.2% ± 4.65% before treatment. Their percentages of IL-1β-positive cells were significantly decreased one month after treatment with stem cell educator therapy (Figure 5B, *p* = 0.016). The data suggested that the chronic inflammation may have caused the cellular stress of PB-IPC, leading to the downregulation of GLUT2 expression on PB-IPC.

## 3. Discussion

PB-IPC normally circulates in human blood. Our previous studies demonstrated that there was high expression of GLUT2 on healthy donor-derived PB-IPC [25]. Current clinical data showed low percentages of GLUT2^+^CD45RO^+^ PB-IPC in prediabetic, recent onset, and longstanding T1D patients, which indicates the novel mechanism involved in the pathogenesis of T1D. Notably, the percentages of GLUT2^+^CD45RO^+^ PB-IPC were dramatically increased in T1D subjects after treatment with stem cell educator therapy. The up-regulated expression of GLUT2 may transport more glucose into the cytoplasm and produce more energy through the mitochondrion-dependent adenosine triphosphate (ATP) production. Due to the high capability of PB-IPC giving rise to insulin-producing cells [24], the restoration of PB-IPC’s function may potentially contribute to the improvement of islet β-cell function.

PB-IPC displays multiple potentials for differentiation in the presence of different inducers, such as neuronal cells and retinal pigmented epithelial (RPE) cells [25,26]. Specifically, PB-IPC exhibits high capability of giving rise to insulin-producing cells and homing to pancreatic islets through its expression of chemokine receptor C-X-C chemokine receptor 4 (CXCR4) and stromal cell-derived factor 1 (SDF-1), which was expressed by pancreatic islets [24]. These biological functions are dependent on the ATP production from mitochondria and coupling with glucose uptake/transferring into cells via GLUT2. Therefore, low or no expression of GLUT2 on PB-IPC resulted in the dysfunction of PB-IPC. Current clinical data demonstrated that the GLUT2 expressions were markedly upregulated on PB-IPC in prediabetic and diabetic patients after the treatment with stem cell educator therapy, leading to the restoration of PB-IPC’s functions and contributing to the tissue repairs in the damaged pancreatic islets.

To date, functional insulin-producing cells have been produced from embryonic stem (ES) cells and induced pluripotent stem cells (iPS) through different protocols of ex vivo induction of differentiations [36,37,38,39,40,41], providing alternative approaches to overcome the shortage of insulin-producing cells. However, in addition to their safety concerns, increasing evidence demonstrates that ES cells, iPS, and their differentiated cells can also cause immune rejection after transplantation [42], limiting their clinical therapeutic potentials. On the contrary, autologous PB-IPC can function as multipotent stem cells, giving rise to different cellular lineages [25,26] and repairing damaged or aged tissues, which may circumvent those issues associated with ES and iPS cells. In this regard, clinical studies demonstrated improvement of islet β cell function in T1D subjects [19] after receiving the treatment with stem cell educator therapy.

Notably, we found the percentages of GLUT2^+^CD45RO^+^ PB-IPC were also very low at baseline in patients with other autoimmune- and inflammation-associated diseases. Their percentages were markedly upregulated post-immune education with stem cell educator therapy. It indicated that there was a common mechanism underlying the pathogenesis of T1D and other autoimmune- and inflammation-associated diseases. It is possibly caused by the high profile of inflammation such as IL-1β in these subjects with T1D and other autoimmune- and inflammation-associated diseases. Clinical data demonstrated that the levels of IL-1β-positive cells were marked decreased in these subjects at first month follow-up after receiving the stem cell educator therapy. In line with this note, Fu et al. reported that Glut2 expression became downregulated in CD8^+^ T cell inflammatory environments [43].

IL-1β is the major inflammatory cytokine released by monocytes/macrophages [44]. Macrophages are professional antigen-presenting cells distributed in almost every organ, such as the brain (named microglia cells), lung (named dust cells), heart, liver (named Kupffer cells), and pancreatic islets [45,46]. Macrophages produce numbers of inflammatory cytokines [47] and chemokines [34], contributing to the polarization and function of macrophages in different diseases [48,49,50,51,52]. Current studies primarily focused on IL-1β with limited numbers of patients having different diseases, which shared the common nature of high levels of IL-1β-positive cells in their peripheral blood samples. It is expected that different clinical disorders may display their disease-specific inflammatory profiles involved in the tissue-specific pathogenesis. Further clinical studies are needed to clarify the networks of these inflammatory cytokines/chemokines and determine the mechanism underlying the inflammation causing the downregulation of GLUT2 expression on PB-IPC. Macrophages are typically subdivided into two subpopulations: type 1 macrophages (M1, pro-inflammatory) and type 2 macrophages (M2, anti-inflammatory) [47]. For decades, T1D has been thought to be due to dendritic cell (DC)-initiated, T-cell-mediated autoimmune destruction of islet β cells [11,53]. However, recent characterization of NOD mice indicated that T-cell dysfunction in T1D is initiated by M1 F4/80^+^CD11c^+^ islet macrophages [54,55,56]. Our previous studies demonstrated the infiltration of F4/80^+^ macrophages against the islet nerves, shedding new light on the pathogenesis of T1D development [57]. IL-1β, produced by monocytes/macrophages, is a master regulator of inflammation via controlling a variety of innate immune processes [58] and involved in the pathogenesis of T1D development [59,60]. Flow cytometry analysis demonstrated the high expression of IL-1β in patients with T1D and other autoimmune- and inflammation-associated diseases. Previous studies revealed that CB-SC could release extracellular vesicles (EV) and turn the macrophages from M1 to M2, which support current clinical data showing the decrease in IL-1β in these patients after treatment with stem cell educator therapy. In line with these T1D clinical data, our previous clinical studies showed that the percentage of monocytes expressing an M1 marker was markedly decreased in type 2 diabetic (T2D) patients four weeks after stem cell educator therapy and that co-culture of CD14^+^ monocytes with CB-SC significantly down-regulated numbers of inflammation-related genes, including chemokines, multiple cytokines, and matrix metallopeptidases [22].

## 4. Materials and Methods

### 4.1. Patients

Patients were enrolled and received treatment with stem cell educator therapy at Throne’s outpatient facility according to the US FDA-approved clinical protocol for type 1 diabetes (IND 19247, ClinicalTrials.gov Identifier: NCT04011020) (Figure 6). The consent form and clinical protocol have been approved by the central Institutional Review Board (IRB) at Advarra (Columbia, MD, USA). The signed consent forms were obtained from all participants. Throne has received Blood Bank License approval for the stem cell educator therapy from the Department of Health in New Jersey. For T1D recruiting, patients were qualified for enrollment if they met the 2022 diagnosis standards of the American Diabetes Association, and a blood test confirmed the presence of at least one autoantibody to pancreatic islet β cells. Patients with other serious/life-threatening autoimmune- and inflammation-associated diseases (e.g., Chonlangitis, Transverse myelitis, Sjogren syndrome, Eczema, Rheumatoid arthritis, and Parkinson’s diseases) received stem cell educator therapy through the Right-to-Try Act program. Patients with alopecia areata received the stem cell educator therapy according to the FDA-approved clinical protocol (IND 19246, ClinicalTrials.gov Identifier: NCT04011748). Exclusion criteria included clinically significant liver, kidney, or heart disease; pregnancy; immunosuppressive medication; viral diseases; or diseases associated with immunodeficiency. Patients were scheduled to perform the follow-ups at first, third, sixth, ninth, and twelfth months after the treatment with stem cell educator therapy.

### 4.2. Culture of CB-SC and Co-Culture of CB-SC with Patients’ PBMC

The culture of CB-SC was performed as previously described [19]. In brief, human umbilical cord blood units were collected from healthy donors and purchased from the Lifeline Stem Cell Tissue/Cord Blood Bank (New Haven, IN, USA). Lifeline Stem Cell Tissue/Cord Blood Bank has received all accreditations for cord blood collections and distributions, with the hospital institutional review board approval and signed consent forms from donors. Mononuclear cells were isolated with Ficoll-hypaque (γ = 1.077, GE Health, Bronx, NY, USA), and red blood cells were lysed using the ammonium-chloride-potassium (ACK) lysis buffer (Lonza, MD, USA). The remaining mononuclear cells were planted in stem cell educator device (Throne Biotechnologies, Paramus, NJ, USA). Cells were cultured in X-VIVO 15 chemically defined serum-free culture medium and incubated at 37 °C with 8% CO_2_ for 14–21 days. When the confluence of CB-SC cultures was more than 80%, they were ready to treat patients’ PBMC. For the co-culture of CB-SC with patients’ PBMC, stem cell educators with CB-SC cultures were initially washed twice with 0.9% sodium chloride and utilized to treat patients’ PBMC isolated by apheresis. After overnight co-culture, the stem cell educator-treated PBMC were collected for quality control (QC) testing and then released for infusion according to the FDA-approved protocol.

### 4.3. Flow Cytometry

Phenotypic characterization of PB-IPC and IL-1β-positive cells was performed by flow cytometry with specific markers including FITC-conjugated anti-human lineage cocktail 1 (Lin1) (CD3, CD14, CD16, CD19, CD20, CD56), PE-conjugated anti-human GLUT2 antibody (R & D Systems, Minneapolis, MN, USA), phycoerythrin-Cy5.5 (PE-Cy5.5)-conjugated mouse anti-human SOX2 (Biolegend, San Diego, CA, USA), PE-Cy7-conjugated mouse anti-human CD45RO (Biolegend, San Diego, CA, USA), APC-Alexa Fluor 750-conjugated mouse anti-human CCR7 (Biolegend, San Diego, CA, USA), Alexa Fluor 647-conjugated mouse anti-human OCT3/4 (Biolegend, San Diego, CA, USA), BV421-conjugated mouse anti-human CD34 (Biolegend, San Diego, CA, USA), BV510-conjugated mouse anti-human leukocyte common antigen CD45 (Biolegend, San Diego, CA, USA), pacific blue (PB)-conjugated mouse anti-human IL-1β (Biolegend, San Diego, CA, USA), FITC-conjugated mouse anti-human interferon γ (Beckman Coulter, Brea, CA, USA), FITC-conjugated mouse anti-human interferon γ (Beckman Coulter, Brea, CA, USA), PE Cy7-conjugated mouse anti-human IL-13 (Biolegend, San Diego, CA, USA), APC-conjugated mouse anti-human IL-17A (Biolegend, San Diego, CA, USA), and pacific blue (PB)-conjugated mouse anti-human tumor necrosis factor (TNF)α (Biolegend, San Diego, CA, USA). Isotype-matched immunoglobulins (IgGs) served as controls. For intra-cellular staining, cells were fixed and permeabilized according to the PerFix-nc kit (Beckman Coulter, Brea, CA, USA) manufacturer’s recommended protocol. After staining, cells were collected and analyzed using a Gallios Flow Cytometer (Beckman Coulter, Brea, CA, USA) equipped with three lasers (488 nm blue, 638 red, and 405 violet lasers) for the concurrent reading of up to 10 colors. The final data were analyzed using the Kaluza Flow Cytometry Analysis Software (Kaluza Analysis 2.1, Beckman Coulter, Brea, CA, USA).

### 4.4. Statistical Analysis

Statistical analysis of data was performed with the GraphPad Prism 10 (version 10.1.0) software. The normality test of samples was evaluated using the Shapiro-Wilk test. Statistical analysis of data was performed using the two-tailed paired *t*-test for the statistical analysis of two groups to determine statistical significance for parametric data between baseline and post-treatment. The Mann-Whitney U test was utilized for non-parametric data. The number “n” in each figure legend represents the biological replicates for clinical studies. Values are given as mean ± SD (standard deviation). Statistical significance was defined as *p* < 0.05.

## 5. Conclusions

Current clinical studies revealed that T1D patients shared the common clinical features with other patients having autoimmune- and inflammation-associated diseases: (1) displaying the low or no expression of GLUT2 on PB-IPC at baseline; and (2) exhibiting a high profile of the inflammatory cytokine IL-1β. Notably, flow cytometry demonstrated that their GLUT2 expressions on PB-IPC were markedly upregulated, and the levels of IL-1β-positive cells were significantly downregulated after the treatment with stem cell educator therapy, leading to the improvement of clinical outcomes.

## Figures and Tables

**Figure 1 ijms-25-08337-f001:**
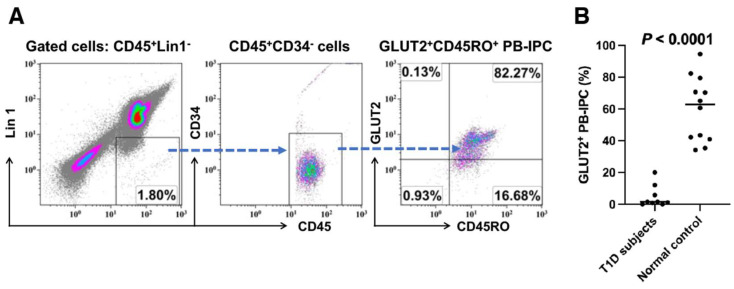
Phenotypic characterization of PB-IPC in human peripheral blood. (**A**) Expression of GLUT2^+^CD45RO^+^ PB-IPC in healthy donors. Representative data from one of twelve preparations. Flow cytometry analysis showed the gating strategy by using the FITC-conjugated anti-human lineage cocktail 1 (Lin1) (CD3, CD14, CD16, CD19, CD20, CD56) to eliminate the known cellular lineages such as T cells, monocytes/macrophages, granulocytes, B cells, and natural killer (NK) cells. Anti-human leukocyte common antigen CD45 was utilized to exclude the red blood cells (RBC) and platelets’ contamination during data analysis. CD34 was applied to discriminate the hematopoietic stem cells. Isotype-matched IgGs served as negative controls. (**B**) Low percentage of GLUT2^+^CD45RO^+^ PB-IPC in recent onset T1D patients (n = 9) in comparison with the healthy control (n = 12).

**Figure 2 ijms-25-08337-f002:**
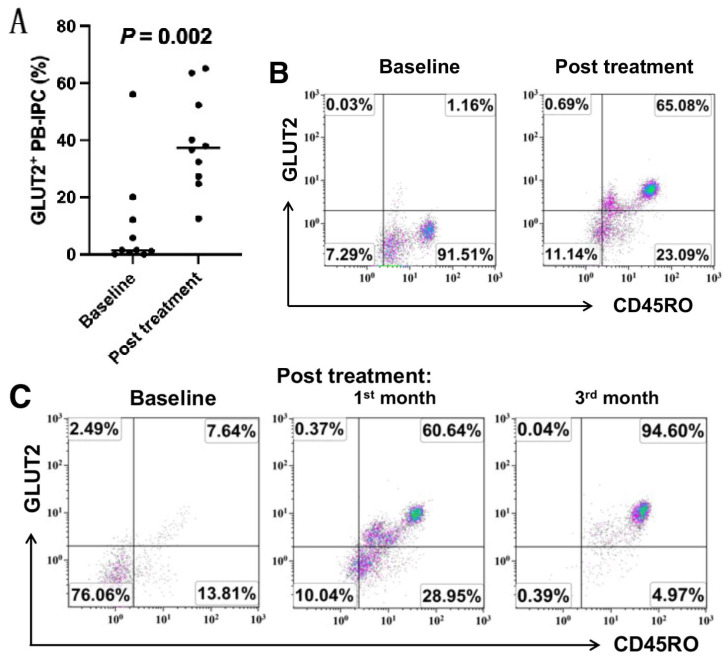
Increase in the percentage of GLUT2^+^CD45RO^+^ PB-IPC in patients after the treatment with stem cell educator therapy. Patients received the stem cell educator therapy according to the FDA-approved clinical protocol (IND #19247) for type 1 diabetes. Flow cytometry analysis was performed at the baseline before and after receiving stem cell educator therapy. The GLUT2^+^CD45RO^+^ PB-IPC were gated from the population of Lin1^−^CD34^−^CD45^+^CD45RO^+^CCR7^+^ cells. Isotype-matched IgGs served as negative controls. (**A**) The percentage of GLUT2^+^CD45RO^+^ PB-IPC was upregulated in T1D subjects one month after stem cell educator therapy. n = 10. The Wilcoxon matched-pairs signed rank test was utilized for the non-parametric data. (**B**) Representative data showed the increased expressions of GLUT2^+^CD45RO^+^ PB-IPC in T1D patient HU2058 before and after the stem cell educator therapy. (**C**) Representative data showed the increased expressions of GLUT2^+^CD45RO^+^ PB-IPC in a prediabetic patient HU2039 before and after the stem cell educator therapy.

**Figure 3 ijms-25-08337-f003:**
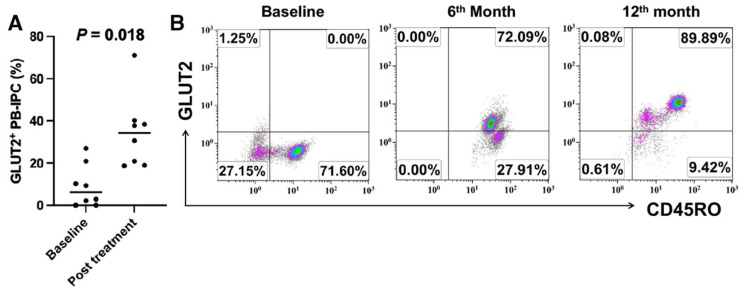
Increase in the percentage of GLUT2^+^CD45RO^+^ PB-IPC in patients with longstanding T1D after the treatment with stem cell educator therapy. (**A**) Markedly upregulated percentages of GLUT2^+^CD45RO^+^ PB-IPC in patients (n = 8) with longstanding T1D after the treatment with stem cell educator therapy for 1 month. (**B**) Representative data from subject HU2003 (diagnosed with T1D for 58 years since age 1 year old) showing improved expression of GLUT2^+^CD45RO^+^ PB-IPC and continued increased level of GLUT2^+^CD45RO^+^ PB-IPC at the twelfth month follow-up.

**Figure 4 ijms-25-08337-f004:**
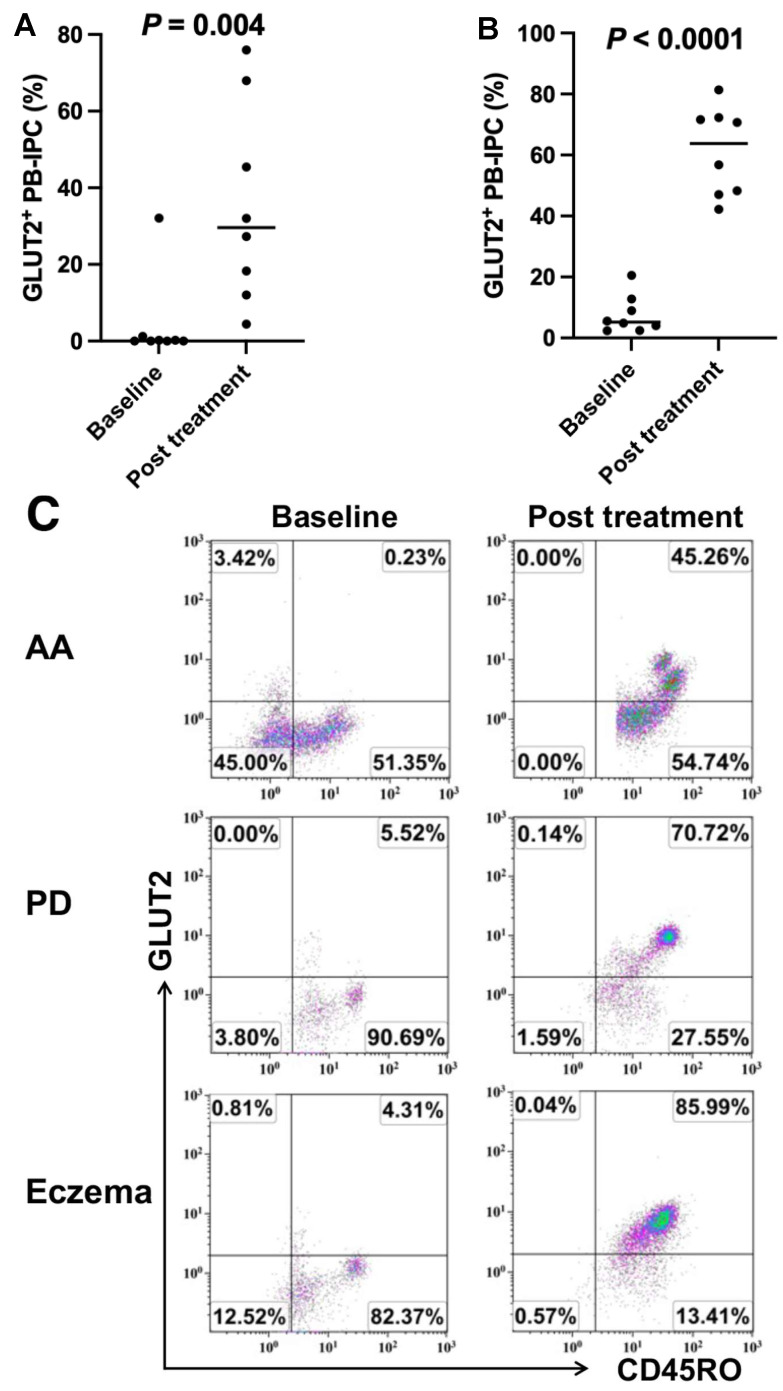
Increased expression of GLUT2 on PB-IPC in other patients after receiving stem cell educator therapy. (**A**) Improvement of the GLUT2 expression on PB-IPC of patients with alopecia areata (AA, n = 8.) after stem cell educator therapy for 3 months. (**B**) Upregulation of the GLUT2 expression on PB-IPC of patients with autoimmune- or inflammation-associated diseases (n = 8) after stem cell educator therapy for 1 month. (**C**) Flow cytometry analysis showing the increased percentages of GLUT2 expression from the baseline levels (left panels) to normal levels post-treatment (right panels) with stem cell educator therapy. Representative data were obtained from subjects HU2027 with AA (top panels), HU2054 with Parkinson’s disease (middle panels), and HU2064 with longstanding Eczema (bottom panels), respectively.

**Figure 5 ijms-25-08337-f005:**
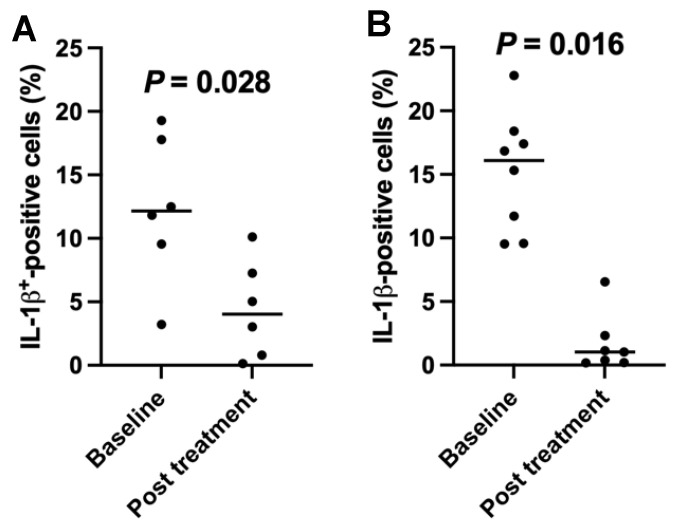
Reduction of the percentages of inflammatory cytokine IL-1β expression in patients after stem cell educator therapy. All subjects received treatment with SCE therapy. Patients’ peripheral blood mononuclear cells (PBMC) were isolated for flow cytometry analyses at baseline and one month after SCE therapy by using the intra-cellular PerFix-nc cellular staining kit (Beckman Coulter, Brea, CA, USA) and followed by analysis in a Gallios Flow Cytometer (Beckman Coulter). Isotype-matched IgGs served as controls. (**A**) Downregulation of the percentages of IL-1β-positive cells in recent onset T1D patients (n = 6) after the treatment with stem cell educator therapy. (**B**) The percentages of IL-1β-positive cells in patients (n = 8) with other autoimmune- and inflammation-associated diseases after treatment with stem cell educator therapy.

**Figure 6 ijms-25-08337-f006:**
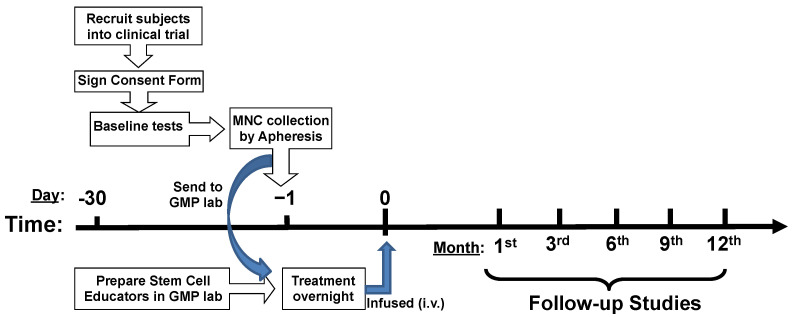
Outline the clinical protocol for stem cell educator therapy and follow-up studies.

## Data Availability

The data that support the findings of this study are available from the corresponding author upon request.

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
