# Peer review of "Increase in the Expression of Glucose Transporter 2 (GLUT2) on the Peripheral Blood Insulin-Producing Cells (PB-IPC) in Type 1 Diabetic Patients after Receiving Stem Cell Educator Therapy"

_ijms, 2024, doi:10.3390/ijms25158337_

Round 1

Reviewer 1 Report

Comments and Suggestions for Authors

Some comments should be carefully addressed as follows:

1. Intro: the consequence of T1D in addition to the available therapeutics strategies should be discussed.

2. Line 84: examined, please correct. I found that the authors only used flow cytometry analysis in this study; thus, I recommend doing mRNA expression for the respective genes. Some values, such as p=0.0037 could be p=0.004. The authors should evaluate other inflammatory biomarkers since IL-1β is not enough to draw the conclusion about the inflammation. For instance, IL-6, TNF-α, and NF-κB, which act as a modulator and a leading signal for inflammatory biomarkers.

3. The authors are encouraged to add a schematic illustration, exhibiting the flow and the design of investigations.

4. Discussion: the limitation of this study should be stated and support this part with future perspectives. Line 252-255: please add evidence for this statement.

5. Methods: how many cells did you culture/wells or flasks? 

Author Response

Many thanks for your kind consideration and comments! Please see our responses in an attachment. 

Reviewer 2 Report

Comments and Suggestions for Authors

In this paper, Zhao and colleagues evaluate the clinical efficacy of Stem Cell Educator therapy in the restoration of islet beta cell function, by exploring the GLUT2 expression on peripheral blood insulin-producing cells (PB-IPC), in recent onset and longstanding T1D patients. For that, the authors isolated peripheral blood mononuclear cells (PBMC) for flow cytometry analysis of PB-IPC and other immune markers before and post the treatment with Stem Cell Educator therapy. The manuscript subject is interesting; however, there are some minor points that could improve the paper. 

Minor points:

1.     It very interesting that the authors found an increase of GLUT2 expression on PB-IPC in recent onset T1D subjects after receiving Stem Cell Educator therapy. But do the authors have any information on the C-peptide levels, glycated hemoglobin A1C values (HbA1C) for all these patients? Was there any impact on their respective insulin doses?

2.     The role of pro-inflammatory cytokines in beta cells is complex. The authors analyzed the expression of several pro-inflammatory cytokines and found differences on the IL1B expression. But knowing that IFN-alpha is a key component of the early stages of T1D, were any differences found in terms of INF-alpha expression?

Author Response

Many thanks for your kind consideration and comments! Please see the responses in an attachment. 

Round 2

Reviewer 1 Report

Comments and Suggestions for Authors

The authors considered the previous claims and explained or responded to them properly. Therefore, from my point of view, the current version could be accepted for publication. 

Author Response

Dear Reviewer,

Many thanks for the reviewer’s kind consideration and comments, which were very helpful guiding us to improve the quality of this manuscript. I appreciate the reviewer accepting my responses and revision of this manuscript. 

Best regards,

Yong Zhao, MD, PhD